# Transcriptome Sequencing Reveals the Mechanism behind Chemically Induced Oral Mucositis in a 3D Cell Culture Model

**DOI:** 10.3390/ijms24055058

**Published:** 2023-03-06

**Authors:** Maria Lambros, Jonathan Moreno, Qinqin Fei, Cyrus Parsa, Robert Orlando, Lindsey Van Haute

**Affiliations:** 1Department of Pharmaceutical Sciences, College of Pharmacy, Western University of Health Sciences, Pomona, CA 91766, USA; 2College of Osteopathic Medicine of the Pacific, Western University of Health Sciences, Pomona, CA 91766, USA; 3NextGenSeek Ltd., Cambridge CB18QY, UK

**Keywords:** oral mucositis, everolimus, mTOR, inhibitor, 3D cell culture model, cornification, glycolysis, sterol biosynthetic process, proinflammatory, transcriptomics, molecular signatures, anticancer agents

## Abstract

Oral mucositis is a common side effect of cancer treatment, and in particular of treatment with the mTORC1 inhibitor everolimus. Current treatment methods are not efficient enough and a better understanding of the causes and mechanisms behind oral mucositis is necessary to find potential therapeutic targets. Here, we treated an organotypic 3D oral mucosal tissue model consisting of human keratinocytes grown on top of human fibroblasts with a high or low dose of everolimus for 40 or 60 h and investigated (1) the effect of everolimus on microscopic sections of the 3D cell culture for evidence of morphologic changes and (2) changes in the transcriptome by high throughput RNA-Seq analysis. We show that the most affected pathways are cornification, cytokine expression, glycolysis, and cell proliferation and we provide further details. This study provides a good resource towards a better understanding of the development of oral mucositis. It gives a detailed overview of the different molecular pathways that are involved in mucositis. This in turn provides information about potential therapeutic targets, which is an important step towards preventing or managing this common side effect of cancer treatment.

## 1. Introduction

Cancer treatments are known to cause many adverse effects. Mucosal lesions of the oral mucosa, also known as oral mucositis, are a common complication of classic anticancer treatments, such as chemotherapy and radiotherapy [1]. Oral mucositis is a common side effect of cancer treatment, in particular treatment with mTORC1 inhibitors such as everolimus [2]. Severe mucositis can be complicated by superinfection, bleeding, and uncontrolled pain, and as a consequence of this, can impact a patient’s quality of life. As a result, this painful condition regularly leads to a dose reduction, interruption of anticancer treatment, or even termination of treatment, limiting the effectiveness of the cancer therapy. Therefore, prevention of therapy-related mucositis is a crucial part of cancer treatment. 

Mucositis develops over five phases [3]. During the initiation phase, immediately after chemotherapy or radiation, reactive oxygen species are formed, causing mucosal damage. Prevention of mucositis must happen before or at the initiation stage. In the second phase, also known as the signaling phase, a stimulation of the NF-kB pathway induces the production of proinflammatory cytokines (such as TNF-α, IL-1β, and IL-6). These and other messenger factors lead to increased cell death and the activation of more cellular signaling pathways, eventually leading to the third phase, the amplification phase. The increased production of proinflammatory cytokines causes a cascade of biological effects, including the breakdown of the epithelium, forming an ulcer. In the fourth stage, these ulcers can be colonized by bacteria, resulting in the clinical manifestation of mucositis. The ulcers are deep and wide, often covered by a pseudo membrane consisting of dead fibrinous cells. The fifth phase is the healing stage, when the integrity of the mucosal layer is restored. In most cases, this happens spontaneously within 2–3 weeks as a result of epithelial proliferation and differentiation. Although the tissue might appear healthy, it remains significantly altered.

Although both chemotherapy and radiotherapy can lead to the formation of oral mucositis, patients treated with mammalian target of rapamycin (mTOR) inhibitors, such as everolimus, are especially vulnerable to mucositis (73.4% of patients, with over 30% of those being severe cases) [4]. Everolimus binds to the FK506 binding protein FKBP12 and selectively inhibits the mammalian target of rapamycin (mTORC1) pathway. mTOR signaling is involved in proliferation, survival, invasion, and angiogenesis, and thus is an important target for anticancer therapy [5].

Currently, there are only a few treatments for oral mucositis. A better understanding of the molecular mechanisms of the development of oral mucositis, in particular during the initiation stage, is crucial towards the development of better treatment to prevent this painful side effect of cancer treatment. Here, using a previously characterized three-dimensional human cell culture model of oral mucositis, consisting of both keratinocytes and fibroblasts [6,7], we study the molecular effect of everolimus treatment on the oral mucosa.

## 2. Results

### 2.1. Histological Assessment of a 3D Oral Mucosal Tissue Model

In this study, we used a previously characterized 3D model of oral tissue. Hematoxylin and eosin staining of the control 3D tissue without everolimus treatment (Figure 1a,b) shows a well-differentiated, multilayered epithelium that resembles the in vivo mucosal epithelium, with a basal layer of cylindrical cells (stratum basalis), the suprabasal stratum spinosum characterized by spinous cells, and the stratum corneum on top. After 60 h of everolimus treatment, the stratum corneum loses its coherence and resembles shredded ribbons (Figure 1c,d).

### 2.2. Treatment with Everolimus for 60 h Affects Cornification

To better understand the side effects of everolimus treatment, we performed RNA-Seq on a previously characterized three-dimensional cell culture model of oral mucositis, consisting of fibroblasts and keratinocytes. Samples were taken after 60 h, treated with 32 or 64 ng everolimus, and compared with untreated control tissue.

The most affected cellular component pathways after 60 h of treatment are epithelial cell differentiation and intermediate filament processes (Figure 2a).

During the terminal differentiation, epidermal keratinocytes form keratin intermediate filaments. This is part of a process known as cornification, which occurs in the upper skin layer. In humans, 54 functional genes exist which code for the different keratin families. Keratins are expressed in a highly specific pattern related to the tissue type and the stage of cellular differentiation (Figure 2b) [8,9]. We analyzed the most differentiated pathways more in depth for all 60 h samples (0 ng, 32 ng, and 64 ng). Many of the proteins of the late cornified envelope (LCE) gene cluster are upregulated at the lower 32 ng dose, but they are slightly downregulated or unchanged at the higher dose of 64 ng compared with the untreated control. Most other genes that are involved in epithelial cell differentiation, intermediate filament organization, and keratinization are altered in a dose-dependent way (Figure 2c–e). 

### 2.3. Treatment with Everolimus for 60 h Affects Proinflammatory Pathways

Inflammatory cell damage and proinflammatory cytokines have been shown to play an important role in the development of oral mucositis. Therefore, we wanted to investigate the RNA levels of these inflammatory pathways. Surprisingly, there were enormous differences in pro-inflammatory cytokines. For example, IL-1B is downregulated in tissues treated with everolimus for 60 h, while IL-6 is strongly upregulated and TNF-α is somewhat upregulated (Figure 3a,b and Appendix A). 

The transcription factor AP-1 plays an important role in wound healing and it is thought to be required under conditions where the balance of keratinocyte proliferation and differentiation has to be rapidly altered, such as during tissue regeneration [10]. Our results confirm that in our oral mucositis model, cells are expressing AP-1, presumably in an attempt to heal the tissue (Figure 3c).

### 2.4. Treatment with Everolimus for 60 h Lowers Glycolysis in the Cells 

Hyperglycemia is one of the major side effects in patients treated with everolimus, affecting 10–50% of all patients [11]. It is known that mTOR is related to glucose transporter (GLUT) expression and it has been shown in mouse muscle cells that everolimus treatment impairs glucose metabolism by lowering the activities of glycolysis and the pentose phosphate pathway, but not the TCA cycle [12].

To study the RNA levels of enzymes involved in the glucose metabolic pathway, we performed RNA-Seq on the 3D oral mucosal tissue model with and without treatment with everolimus (60 h). Looking at the most differentially expressed genes, it becomes immediately clear that glycolysis is strongly reduced, as several of the most important enzymes, such as PFKM, ALDOA/C, TP1, and PGK1, have strongly reduced RNA levels at the higher dose of 64 ng, as opposed to mildly reduced levels when treated with 32 ng (Figure 4a,b and Appendix A). 

**Figure 2 ijms-24-05058-f002:**
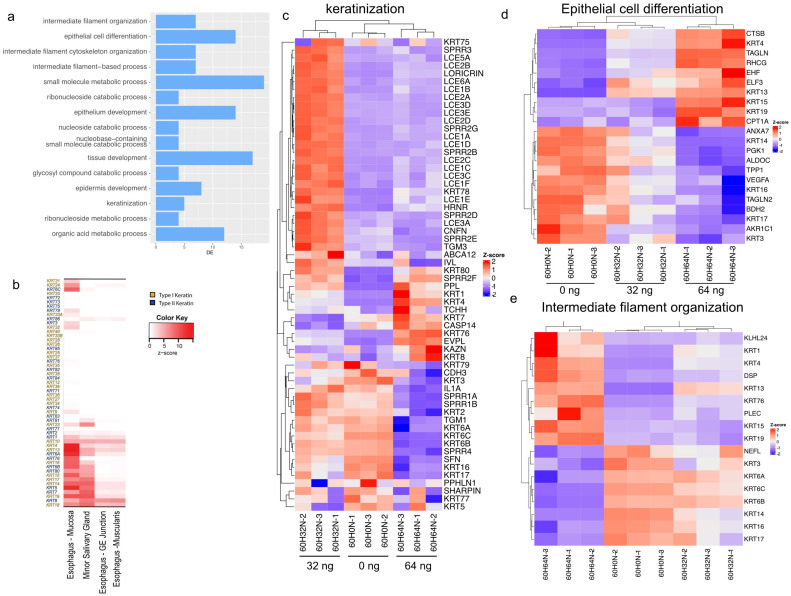
Effect of everolimus treatment on cell differentiation. (**a**) GO enrichment analysis for biological processes showing the fifteen most significantly changed pathways sorted according to significance (*y*-axis). The number of differentially expressed genes is shown on the *x*-axis. (**b**) Selection of tissue-specific keratin expression in adult tissues (adapted from Ho et al., 2022 [13]). (**c**) Heatmap illustrating RNA-Seq differential expression data for genes associated with keratinization (GO:0031424). (**d**) Heatmap showing differential expression data for genes associated with epithelial cell differentiation (GO:0030855, padj < 0.01). (**e**) Heatmap showing the RNA-Seq results for genes associated with intermediate filament organization (GO:0045109, padj < 0.01).

**Figure 3 ijms-24-05058-f003:**
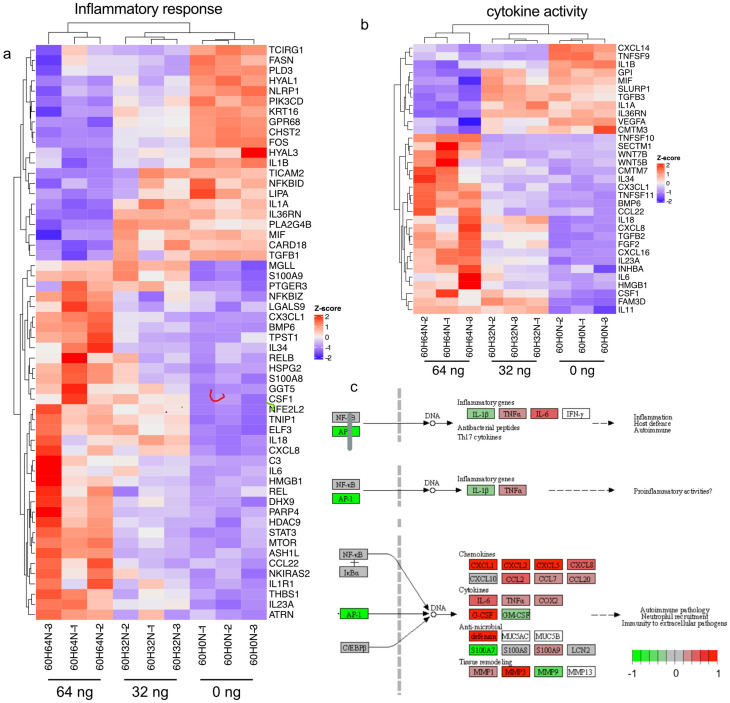
Effect of everolimus treatment on proinflammatory cytokines. (**a**) Heatmap illustrating RNA-Seq differential expression for genes associated with inflammatory response (GO:0006954, padj < 0.05). GO enrichment analysis for biological processes. (**b**) Heatmap illustrating RNA-Seq differential expression data for genes associated with cytokine activity (GO:0005125, padj < 0.01). (**c**) Pathview image showing a selection of inflammation and cytokines (in tissue treated with 64 ng everolimus compared with untreated after 60 h) (green means downregulated and red means upregulated). (Appendix A is the full figure.)

FOXK1 is involved in mTORC1-mediated metabolic reprogramming. In response to mTORC1 signaling, FOXK1 regulates the expression of genes associated with glycolysis and downstream anabolic pathways, thereby regulating glucose metabolism. In the tissue treated with 64 ng of everolimus, FOXK1 is strongly upregulated, suggesting that this enzyme plays an important role in the downregulation of glycolysis in these samples (Appendix A).

**Figure 4 ijms-24-05058-f004:**
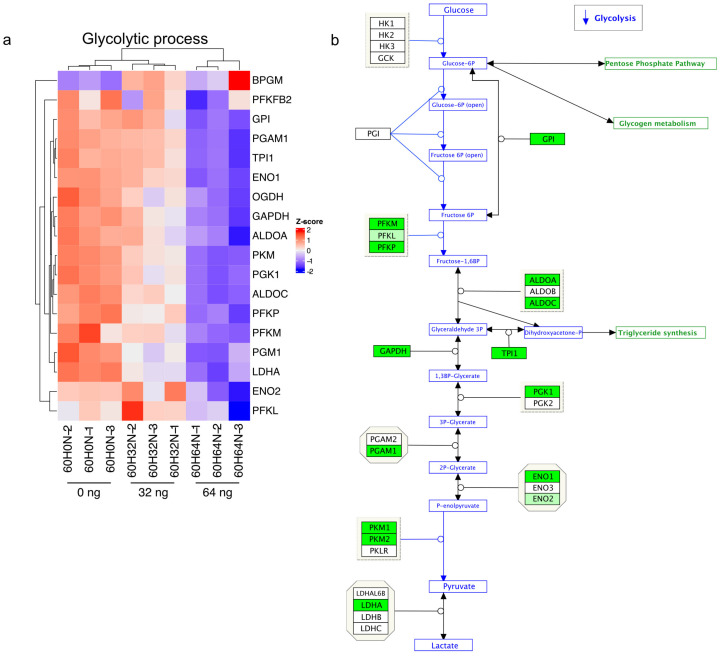
Effect of everolimus treatment on glycolysis. (**a**) Heatmap illustrating RNA-Seq differential expression for genes associated with glycolytic process (GO:0006096, padj < 0.5). (**b**) Schematic representation of the glycolysis pathway with the genes that are significantly overexpressed in everolimus-treated cells compared with untreated tissue after 60 h indicated in green (and slightly upregulated in light green).

### 2.5. Treatment with Everolimus for 60 h Affects Cell Cycle Control and Cell Division 

Everolimus is known to cause induction of autophagy and G1 cell cycle arrest [14], so we wanted to see how this is reflected in our oral mucositis model. We observed a strong reduction in aurora kinase A (AURKA), a cell cycle-regulated kinase that appears to be involved in microtubule formation and stabilization at the spindle pole during chromosome segregation (Figure 5a) [15]. Genes that are associated with microtubules, such as TUBG2, TUBA1B, TUBB2B, TUBB2A, and TUBG1, are also strongly downregulated, presumably as the result of the lower AURKA expression. Furthermore, several centromere proteins, such as CENPF [16] and CENPT [17], that bind microtubules are downregulated as well.

We also looked at the effect of everolimus treatment on autophagy. Inhibition of mTORC1 leads to an increased expression of RNA levels of RB1-inducible coiled coil 1 (RB1CC1), which plays an important role in autophagosome formation and further orchestrates the recruitment of autophagy-related (ATG) proteins such as ATG2B, which was also upregulated in our study [18].

### 2.6. Sterol Biosynthetic Process Affected after 40 h of Everolimus Treatment

Next, we wanted to investigate the effect of a shorter treatment time (40 h instead of 60 h). We used the 3D oral mucosa model as described above (Figure 6a,b). A GO enrichment analysis of differentially expressed genes between the non-treated group and the 40 h everolimus treatment group shows that the sterol biosynthetic process is the most affected (Figure 6c). A more detailed look into the genes involved in this process shows that most differentially expressed genes in this pathway are downregulated with everolimus treatment. This consists mostly of genes involved in the cholesterol synthesis pathway (Figure 6e), such as mevalonate kinase (MVK) and HMG-coA reductase (HMGCR). Proper regulation of sterol biosynthesis is crucial for cellular homeostasis and a dysregulated sterol metabolism engages pathways that lead to inflammation [19].

## 3. Discussion

The inflammation response of epithelial mucosa to everolimus cancer treatment leads to mucositis, a painful side effect, often causing an interruption or discontinuation of treatment. Patients who develop mucositis have four times the risk of death compared with patients who do not develop this common side effect [20]. Preventive measures such as good oral hygiene and a dental assessment are important, but cannot fully prevent oral mucositis. Analgesics can provide some pain control. Patients are also given antifungals, steroids, and antiseptics. Dexamethasone mouthwashes are effective at reducing the incidence and severity of mucositis [21], but it is still crucial to develop even better treatments. This might be possible with a better understanding of the molecular mechanism of mucositis. Here, we show a detailed overview of the changes in RNA levels during the development of oral mucositis using a previously characterized 3D tissue model [6]. 

We show that the most affected pathways are cornification, cytokine expression, glycolysis, and cell proliferation.

The maintenance of the mature epidermis relies on a tightly balanced process of keratinocyte proliferation and terminal differentiation mTOR inhibitors, such as sacrolimus and to a lesser degree everolimus, have been shown to cause problems in wound healing in transplant patients [22,23]. In our study, after everolimus treatment, the expression levels of keratins were opposite of what normally occurs during wound healing [24]. There was an upregulation of KRT1 and KRT4 and a downregulation of KRT16/17 and KRT6. We know that KRT4 is associated with the basal layer [8].

Several studies [24,25,26] have established the important role of the mTOR1 complex in regulating the innate immune response, and the mTOR pathway is targeted as an immunosuppressive and anti-proliferative therapy. However, during the development of mucositis, a reduction in mTOR activity stimulates the translation of more proinflammatory cytokines, eventually leading to a cascade of processes and the development of oral ulcers. To be able to prevent mucositis, a better understanding of the cytokines involved in this process is crucial. In this study, we show how everolimus treatment affects the RNA levels of cytokines and cytokine receptors (Appendix A) and we show that transcription factor AP-1, which plays an important role in wound healing, is downregulated in oral mucosal tissue treated with 64 ng everolimus (Figure 3c). Our results show that downregulation of AP-1 plays an important role in the development of mucositis by stimulating proinflammatory cytokines and chemokines such as IL-6 [13].

It also has been shown that mTORC1 stimulates glucose uptake and enhances mitochondrial biogenesis through activation of PPARγ coactivator 1α (PGC1α). Furthermore, in a genetic knockout model in which mTORC1 is activated, glycolysis, mitochondrial respiration, and lipid synthesis were upregulated [27]. In this study, we show that inhibition of mTORC1 by everolimus is responsible for a downregulation of most genes that are involved in glycolysis in a dose-dependent manner (Figure 4a,b). Interestingly, sterol synthesis is downregulated in a time-dependent manner (Figure 6c–e).

Everolimus selectively inhibits the mTORC1 pathway, which is also involved in proliferation, cell growth, and autophagy. In this study, we observed a strong reduction in aurora kinase A (AURKA) and genes associated with microtubules (Figure 5a).

In sarcomatoid metastatic renal cell carcinoma, AURKA levels were increased, resulting in an increased mTOR pathway [28], confirming the close interaction of AURKA and mTORC1. AURKA activity is associated with drug resistance in breast cancer. Suppression of AURKA induces sensitivity to everolimus by inducing cell death in vivo [29], again confirming the close association between the mTORC1 pathway and AURKA, suggesting that AURKA plays an important role during the development of mucositis.

Samples taken at an earlier timepoint (40 h instead of 60 h) show a strong alteration in genes involved in the sterol biosynthetic process, providing a clue about the processes involved earlier in the development of mucositis. It is believed that oxidative stress due to the generation of reactive oxygen species plays a crucial role in the initiation of chemoradiation-induced oral mucositis [30]. However, further experiments with sampling at different timepoints within the first 24 h would be necessary to study this process and the role of mitochondria (the main source of reactive oxygen species in both health and disease) in this process.

## 4. Materials and Methods

### 4.1. 3D Oral Tissue Model and Treatment

The tissue used for this experimentation is a validated organotypic co-culture model consisting of normal human keratinocytes grown on top of fibroblasts in serum free media forming a 3D, highly differentiated, multilayer tissue which is histologically very similar to buccal mucosa. The 3D tissue, EpiOral kit, full thickness, was purchased from MatTek Corporation, Ashland, MA. The EpiOral 3D tissue was cultured in cell culture inserts with microporous bottom membranes with a pore size of 0.4 um. The growth medium was placed beneath the culture inserts and the tissue was nourished by media permeating through the microporous membrane. The 3D tissues were treated with 32 and 64 ng/mL of everolimus mixed with the growth medium for 60 h. The treatments were performed in triplicate. Next, the tissues were collected and some were placed in formalin for histopathologic studies; others were used for the extraction of total RNA.

### 4.2. Microscopy

After treatment with everolimus, the 3D tissues were fixed in 10% neutral buffered formalin (Sigma-Aldrich). The formalin-fixed 3D tissues were placed in paraffin (wax) (Sigma-Aldrich) to create formalin-fixed paraffin embedded (FFPE) blocks which then were cut using a microtome and mounted on glass microscope slides. Subsequently, the tissues were processed for staining with hematoxylin and eosin (H and E) and evaluated microscopically to assess the effect of everolimus on the 3D tissues. All images were taken at 40× magnification by the Nikon Digital Sight camera system DS-R11.

### 4.3. RNA Extraction 

Total RNA was extracted using an RNeasy Plus Mini Kit (Qiagen) following the manufacturer’s instructions. RNA integrity was checked by an Agilent Bioanalyzer 2100 (Agilent) and only samples with clean rRNA peaks were used for RNA library preparation.

### 4.4. RNA-Seq Library Preparation

Total RNA (250 ng) was used as input for RNA library preparation using the KAPA Stranded mRNA-Seq (Roche), which includes poly(a) selection. The final library was assessed by an Agilent Bioanalyzer 2100 and a Qubit3.0 Fluorometer (Life Technologies Corp). High throughput sequencing was performed on an Illumina HiSeq 4000 platform (150 bp paired end).

### 4.5. RNA-Seq Analysis

After adapter and quality trimming with TrimGalore! 0.6.5, reads of 20 nucleotides or longer were pseudomapped to the human transcriptome (hg38) using Salmon 1.9.0 [31]. Gene counts were normalized and analyzed for differential expression using DESeq 2.0 [32]. Go.db 3.16.0 [33] and path view 1.38.0 [34] were used for GO pathway analysis and visualization, and complexHeatmap 2.14.0 [35] was used to generate heatmaps.

## 5. Conclusions

The transcriptome analysis of a 3D cell culture model of human oral mucositis treated with everolimus shows that the pathways affected are cornification, cytokine expression, glycolysis, and cell proliferation. This negatively affects wound healing and leads to an inflammation cascade. 

Several studies in the past have studied the effect of mTORC1 inhibition, but they usually have focused on a selection of genes, not the whole transcriptome [23,36]. Here, we provide a thorough whole transcriptome analysis of the effect of everolimus on mTORC1 inhibition in the oral mucosa and the development of oral mucositis. This study provides important information about the exact molecular effect within the cells and consequently determines potential targets for the treatment of oral mucositis.

## Figures and Tables

**Figure 1 ijms-24-05058-f001:**
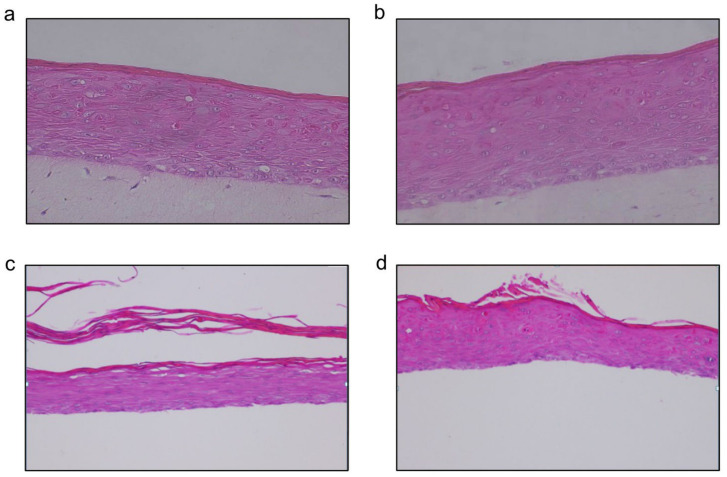
3D human oral tissue model, untreated and treated with everolimus and stained with H and E at different time points. The pink color shows the epithelium which consists of keratinocytes. The top part of the epithelium is the stratum corneum (flat horizontal cells), followed by the stratum spinosum and stratum basalis at the bottom of the epithelium. (**a**) H and E staining of an untreated oral mucosa model (24 h), (**b**) H and E staining of an untreated oral mucosal tissue model (60 h), (**c**) H and E staining of an oral mucosal tissue model treated with 32 ng/mL everolimus (60 h), and (**d**) H and E staining of an oral mucosal tissue model treated with 64 ng/mL everolimus (60 h). The magnification factor for (**a**–**d**) was 40×.

**Figure 5 ijms-24-05058-f005:**
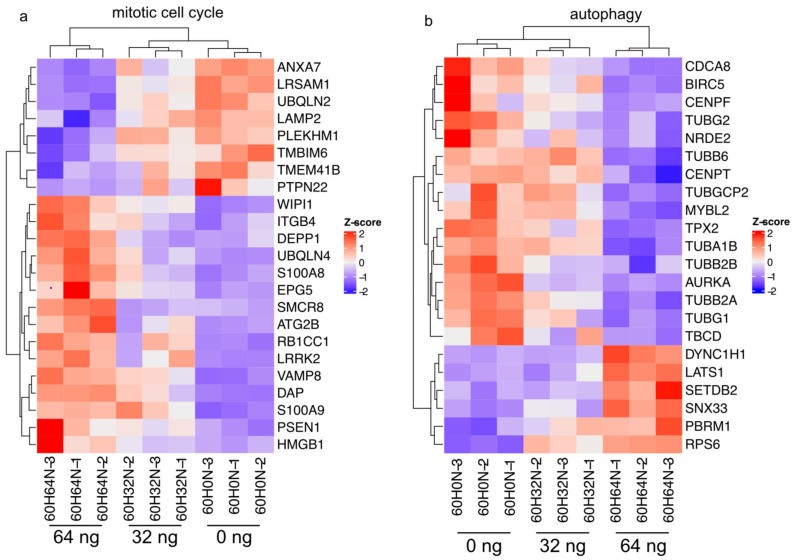
Effect of everolimus treatment on cell cycle and autophagy. (**a**) Heatmap illustrating RNA-Seq differential expression for genes associated with mitotic cell cycle (GO:0000278, padj < 0.01). (**b**) Heatmap illustrating RNA-Seq differential expression for genes associated with autophagy (GO:0006914, padj < 0.01) after 60 h.

**Figure 6 ijms-24-05058-f006:**
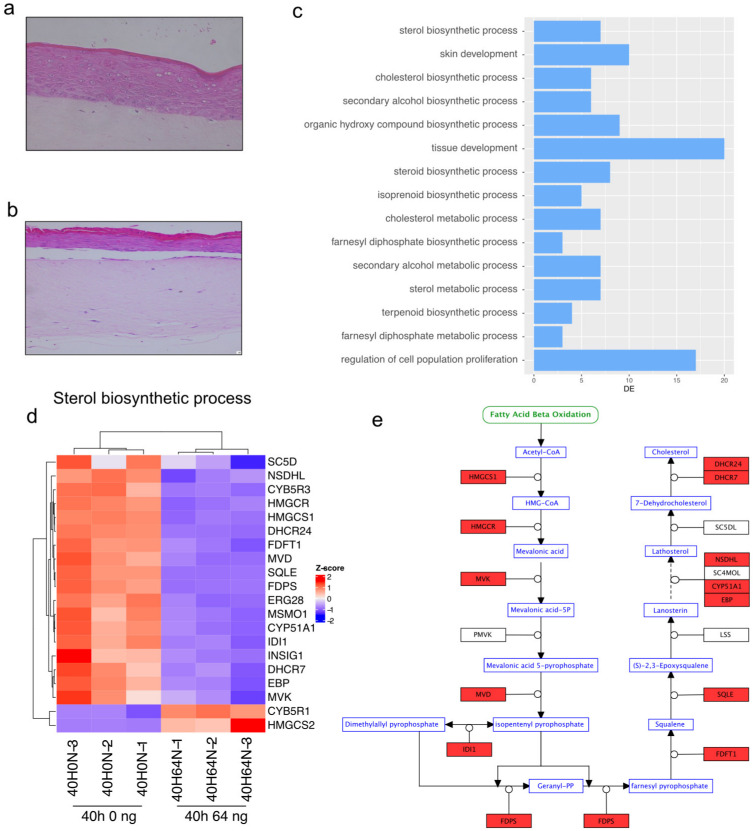
Effect of everolimus treatment for 40 h. (**a**) Hematoxylin and eosin (H and E) staining of an oral mucosa model after 40 h. (**b**) H and E staining of an oral mucosa tissue model treated with 64 ng/mL everolimus for 40 h. The magnification factor for (**a**,**b**) was 40×. (**c**) GO enrichment analysis for biological processes of genes differentially expressed between untreated and tissue treated with 64 ng/mL everolimus for 40 h. Terms are ordered according to significance (*y*-axis) and the number of differentially expressed genes in each process is shown on the *x*-axis. (**d**) Heatmap illustrating RNA-Seq differential expression for genes associated with the sterol biosynthetic process (GO:0016126, padj < 0.05). (**e**) Schematic representation of the cholesterol synthesis pathway with the genes that are downregulated after 40 h of treatment with everolimus in (**d**) in red.

## Data Availability

High throughput sequencing data can be accessed at the GEO data repository (GSE213650).

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
