# Peer review of "Transcriptome Sequencing Reveals the Mechanism behind Chemically Induced Oral Mucositis in a 3D Cell Culture Model"

_ijms, 2023, doi:10.3390/ijms24055058_

Round 1

Reviewer 1 Report

This study explores the molecular mechanism behind the development of oral mucositis, with a focus on the critical initiation stage, in an effort to control chemotherapy side effects. The report is well-structured and effectively presented. My minor comments include:

1.    Please make sure the source of materials is addressed in manuscript perfectly for reproduction of study, for instance: line 84 “the 3D tissues were placed in 10% formalin. The tissues were then stained with he-matoxylin and eosin (H and E) and evaluated microscopically to assess the effect of everolimus on the 3D tissues”

2.    Line 84, what kind of microscopy you used? Please mention it shortly.

3.    Text organization disruption. Revisit lines 88-91.

4.    Referring to my comment 1, see lines 95-96

5.    Line 89, “…After adapter and quality trimming with TrimGalore!? 20nt ?

6.    The conclusion part is missed, it is advised to shortly describe to most important findings of works in that section.

7.    As the time has been investigated as an influential parameter, if possible could you comment on the process of further time reduction to < 10-15 hours

8.    In Figure 1, briefly introduce the dominant colors found in the figure that what they are referring to. 

Reviewer 2 Report

The manuscript by Lambros et al. investigates the changes in the transcriptome in an organotypic 3D oral mucosal tissue model under everolimus treatment. The functional enrichment analysis of differentially expressed genes highlights key pathways crucial for oral mucositis development. This study holds great biomedical significance and will contribute to our understanding of chemically induced oral mucositis. The manuscript is well structured and clearly written, and the results provide strong support for the conclusions. However, the following points could further strengthen the manuscript:

(1) Detailed information regarding the analysis parameters should be provided, such as the version and specific parameters of the bioinformatics tools used (Line 97-102) and the source (e.g. GENCODE, RefSeq, Ensemble) and version of the human gene annotation library. This will assist future researchers in repeating the results.

(2) The figures could be improved by clearly specifying important information in the figure text and/or legend, such as the scale bar for Fig 1a-d and Fig 6a-b, and the y-axis label for Fig 2a and Fig 6c.
